# Topological Photonic Crystal Sensors: Fundamental Principles, Recent Advances, and Emerging Applications

**DOI:** 10.3390/s25051455

**Published:** 2025-02-27

**Authors:** Israa Abood, Sayed El. Soliman, Wenlong He, Zhengbiao Ouyang

**Affiliations:** 1THz Technology Laboratory, Shenzhen Key Laboratory of Micro-Nano Photonic Information Technology, Key Laboratory of Optoelectronic Devices and Systems of Ministry of Education and Guangdong Province, THz Technical Research Center of Shenzhen University, Shenzhen 518060, China; israa_abood@yahoo.com; 2College of Physics and Optoelectronic Engineering, Shenzhen University, Shenzhen 518060, China; 3Physics Department, Faculty of Science, Assiut University, Assiut 71516, Egypt; selshahat@aun.edu.eg; 4College of Electronics and Information Engineering, Shenzhen University, Shenzhen 518060, China; wenlong.he@szu.edu.cn

**Keywords:** topological photonic sensors, topological states, quasi-BICs, sensitivity

## Abstract

Topological photonic sensors have emerged as a breakthrough in modern optical sensing by integrating topological protection and light confinement mechanisms such as topological states, quasi-bound states in the continuum (quasi-BICs), and Tamm plasmon polaritons (TPPs). These devices exhibit exceptional sensitivity and high-*Q* resonances, making them ideal for high-precision environmental monitoring, biomedical diagnostics, and industrial sensing applications. This review explores the foundational physics and diverse sensor architectures, from refractive index sensors and biosensors to gas and thermal sensors, emphasizing their working principles and performance metrics. We further examine the challenges of achieving ultrahigh-*Q* operation in practical devices, limitations in multiparameter sensing, and design complexity. We propose physics-driven solutions to overcome these barriers, such as integrating Weyl semimetals, graphene-based heterostructures, and non-Hermitian photonic systems. This comparative study highlights the transformative impact of topological photonic sensors in achieving ultra-sensitive detection across multiple fields.

## 1. Introduction

Sensors are needed devices paramount in numerous applications, ranging from environmental monitoring and biomedical diagnostics to chemical analysis [1,2,3,4,5]. These devices perceive changes in the physical environment via light interactions, enabling precise measurements of phenomena such as the light intensity, temperature, pressure, and chemical composition. Conventional sensors, which rely on traditional photonic structures, have been widely used [6,7,8,9,10,11,12,13,14]. Nonetheless, the sensitivity to defects, noise interference, and limitations in performance encountered at high precision levels have restrained their development in real-world applications. Recent advancements in photonics have spurred significant innovation in the design and functionality of sensors, with the integration of topological photonics emerging as a breakthrough [15,16,17,18,19,20,21,22,23,24,25,26,27,28,29,30].

The foundations of topological photonics emerged from studying topological insulators (TIs) within solid-state physics [16,19,20,21,22,24,29,31,32]. TI materials are characterized by insulating bulk modes and conducting surface states or modes immune to backscattering, even with impurities or imperfections [31,33,34,35,36,37,38]. This discovery, first exemplified by the integer quantum Hall effect, led to the realization that these topological concepts could be translated into photonic systems. Pioneering work by Haldane and Raghu in 2008 [39,40] extended the ideas of TIs to photonic crystals (PCs), incorporating time-reversal-breaking magneto-optical elements to create photonic bands with nontrivial topological invariants. Their work led to the theoretical prediction of robust, chiral edge states or modes that propagate along the boundaries of these materials. The experimental display of these edge modes in 2009 [41] indicated a critical milestone, setting the foundation for the burgeoning research in topological photonics. Over the years, researchers have extended topological photonics to embrace numerous material platforms, such as plasmonic structures, metamaterials, and even acoustic systems [4,25,42,43,44,45]. These systems offer unique light manipulation with suitable behavior immune to imperfections, promising breakthroughs in advancing high-performance topological sensors [2,46,47,48].

The key distinction between conventional optical and topological photonic sensors lies in the topological protection offered by the latter [49,50,51,52,53]. In traditional optical sensors, light is guided through waveguides, fibers, or resonators, and their behavior is subject to perturbations such as material defects, the surface roughness, or external noise [54]. These imperfections often lead to losses in sensitivity or reliability. On the other hand, the hallmark of topological photonic systems is their robust edge modes—light modes confined to the edges or surfaces of materials [55,56,57]. These modes are immune to backscattering, defects, and imperfections, allowing light to propagate unimpeded along designated paths with minimal loss and ensuring high sensitivity and stability. This unique property makes topological photonic sensors exceptionally well suited for applications requiring high precision and reliability in challenging conditions.

This review focuses on exploring the unique properties of topological photonic sensors, emphasizing their principles, the role of topological protection, and their practical applications. By understanding how the physics of TIs translates into sensor technologies, we aim to showcase the potential of topological photonic sensors to surpass the aptitudes of conventional optical sensors, particularly in fields that need precision, robustness, and resilience in complex environments. The advancement of these sensors signifies a promising focus for improving the reliability and accuracy of optical sensing technologies, inducing pushing the boundaries of what is achievable in photonics.

## 2. Principles of Topological Photonic Sensors

Topological photonic sensors leverage TI properties, demonstrating robust edge modes that are immune to scattering and defects, protected by topological invariants [19,46,58,59,60,61,62,63,64,65,66,67,68,69,70]. The key topological invariant that controls the light behavior in photonic materials is the Chern number (or winding number), which is exploited to characterize the topological phase of a material [18,71]. These invariants are critical because they define the topologically protected edge modes and bulk–edge correspondence, which are central to the operation of topological photonic sensors. The Chern number (C) is an integer representing a material’s topological class [72,73,74,75,76,77,78]. It is derived from the Berry curvature ($B\left(k\right)$), which characterizes the phase properties of wavefunctions in momentum space. The Chern number can be computed as an integral over the entire Brillouin zone (*k*-space) using the following equation [16,31,77,79]:(1)$$C_{n}=\frac{1}{2\pi }\int_{BZ}B\left(k\right)d^{2}k=\frac{1}{2\pi }\int_{BZ}\nabla_{k}\times A^{n}\left(k\right)d^{2}k$$
where $B\left(k\right)$ is the Berry curvature, which measures the geometric phase accumulated by the wave packet as it traverses the momentum space. $A^{n}\left(k\right)=i\left\langle u_{nk}|\nabla_{k}|u_{nk}\right\rangle$ is the Berry connection, which determines the rate at which the geometric phase accumulates, and $u_{nk}$ is the periodic part of the Bloch function of eigenmodes in the nth bulk band. In this context, the presence of i in the Berry connection reflects the phase relationship between the eigenstates of the system, which is a core component in defining geometric and topological phases. The imaginary unit ensures that the Berry connection $A^{n}\left(k\right)$ is a complex-valued vector field, and its curl, the Berry curvature, governs the topological properties of the band structure. This quantity, calculated for each band in the band structure, reflects the overall “twist” of the wavefunctions over the Brillouin zone (BZ) (the overall “twist” of the bands in the deformed band structure). A nonzero $C_{n}$ implies a topologically nontrivial phase, suggesting that edge modes will exist at the boundary between a material with a nonzero $C_{n}$ and a trivial insulator [72,80,81,82]. These edge modes are unidirectional and immune to backscattering, making them robust to imperfections and defects, which is critical for observing sensors in noisy environments. The bulk–edge correspondence principle states that the topological properties of a material’s bulk (described by the Chern number) directly dictate the existence of edge states. When a material with a nonzero Chern number (topologically nontrivial) is in contact with a material with a trivial topological phase (Chern number of zero), chiral edge states will exist at the interface [72,77]. These edge states are robust to scattering and defects, a property that is essential for high stability and reliability for various sensing applications.

In a conventional photonic waveguide, light is guided through the bulk material, and the performance is susceptible to scattering caused by imperfections, leading to a loss of signal strength and degraded sensitivity [83,84,85,86]. In the context of topological photonic sensors, these robust edge states are leveraged to guide light in a way that is not only highly efficient but also resistant to disruption. For example, topological PCs or metamaterials can be designed to create waveguides, resonators, and cavities that support edge modes—paths along the edges of the material where light can travel with minimal loss or scattering [43,87]. The edge modes that arise are one-way (unidirectional) and exist within the bandgap of the bulk material, where there are no other propagating modes. These modes are crucial for ensuring high signal integrity in sensors, particularly in noisy or imperfect environments where traditional systems might fail due to defects in the material or the misalignment of the components. The sensor’s sensitivity comes from the ability of these edge modes to respond to specific changes in the environmental factors—such as variations in the refractive index, temperature changes, changes in chemical concentrations, or pressure fluctuations—with higher precision than conventional sensors. When an environmental parameter changes, the edge mode’s resonant frequency or propagation characteristics shift. The shift can be detected by measuring the change in the transmission, reflection, or phase of the guided light, and this change can be quantitatively related to the environmental parameter. The edge modes are confined to the material’s boundaries, where their strong interaction with the surrounding environment enhances sensing.

In sensor devices, the sensitivity (*S*), the quality factor (*Q*), and the figure of merit (*FOM*) are vital parameters for evaluating the ability and performance of the sensor. One of the key parameters is the sensitivity equation, which quantifies the sensor’s response to external changes and describes the interaction between the edge modes and the environment and perturbation due to external fields. A general definition of sensitivity is given as follows:(2)$$S=\frac{∆\lambda }{∆x},$$
where ∆λ represents the measurable change in the optical property, such as the wavelength change of the edge mode and the change in the transmission, reflection, or phase of the guided light due to the change in the surrounding environment ∆x. ∆x denotes the change in the environmental parameter being measured for specific sensing applications, which can vary depending on the type of sensor, such as the refractive index (∆n), temperature (Δ*T*), or pressure (Δ*P*). It provides a unified framework to calculate sensitivity across different sensor types. Since topological photonic sensors localize light along the edges, a small value of ∆x significantly enhances sensitivity, allowing for the detection of minute environmental changes. In the following, we limit λ as the wavelength of the edge modes. By detecting and plotting the resonance wavelength shifting ∆λ as a function of ∆x variation, the sensitivity can be detected from the linear fitting. Conspicuously, the Q factor of the topological edge mode, a measure of the resonance’s sharpness and defined as the ratio of the resonance frequency ($\lambda_{c}$) to the resonance linewidth of the full width at half maximum (Δλ), can be expressed as [88](3)$$Q=\frac{\lambda_{c}}{\Delta \lambda },$$

The FOM can be defined as the ratio of S to Δλ [11]:(4)$$FOM=\frac{S}{FWHM},$$

The detection limit (DL), representing the most minor detectable change in environmental media, is another critical parameter, calculated as [66](5)$$DL=\frac{\lambda_{c}}{SQ}.$$

Topological photonic sensors represent a powerful integration of advanced material science and photonic engineering. Their underlying principles—topological protection, robustness against defects, and enhanced sensitivity—make them highly suited for next-generation sensing technologies. By tailoring the geometry of the topological material or by varying its intrinsic properties, it is possible to design highly optimized sensors for specific applications. This design flexibility, combined with intrinsic topological protection, allows topological photonic sensors to outperform traditional sensors in terms of stability, sensitivity, and reliability in fields like chemical sensing, environmental monitoring, and medical diagnostics.

The performance of topological photonic sensors in noisy environments is often quantified by the Signal-to-Noise Ratio (SNR), which measures the ratio of the desired signal (e.g., the resonance shift due to a change in the refractive index) to the background noise [89]. A high SNR indicates that the sensor can reliably detect small changes in the target parameter, even in the presence of environmental noise. The robustness of these sensors comes from their topologically protected edge states, which are inherently less susceptible to noise from environmental factors such as material imperfections, temperature fluctuations, and vibration. The high *Q*-factor often achieved in topological photonic sensors leads to a narrower resonance linewidth, increasing the sensor’s ability to resolve small changes in the refractive index, thus improving the SNR. To achieve a better SNR, a cross-polarization technique is applied [90,91].

## 3. Types and Applications of Topological Photonic Sensors

The rapid advancements in topological photonics have paved the way for diverse sensor technologies, each tailored to specific applications in fields ranging from environmental monitoring to biomedical diagnostics [4,92]. Topological photonic sensors offer unparalleled sensitivity, precision, and stability by leveraging unique properties such as topological protection, high-*Q* resonances, and robust topological modes. This section explores the significant types of topological photonic sensors, including refractive index (RI) sensors, biosensors, gas sensors, thermal sensors, and multifunctional tunable designs. Each sensor type is discussed in terms of its working principles, physical insights, and applications, illustrating how topological photonics is transforming the landscape of modern sensing technologies.

### 3.1. Refractive Index Sensors

Refractive index (RI) sensors exploit the interaction of light with the surrounding medium to detect minute changes in the refractive index, enabling high-precision environmental sensing [93,94]. The integration of topological properties enhances these devices by confining light to robust topological states/modes (including edge states/modes, TESs/TEMs, and corner states/modes, TCSs/TCMs), Tamm plasmon polaritons (TPPs), or quasi-bound states in the continuum (quasi-BICs). The robustness of the topological edge states (TESs) in these sensors is guaranteed by the nonzero Chern number, which ensures the unidirectional propagation of light along the material boundaries. As discussed in Section 2, this topological protection is crucial for maintaining high sensitivity and stability in the presence of defects or environmental noise, offering advanced performance metrics in terms of sensitivity and *Q*-factors. For instance, Elshahat et al. [95] investigated the first heterostructure of a one-dimensional (1D) topological PC mirror, leveraging hybrid resonance modes formed at the interface of topologically distinct PCs (see Figure 1a). By introducing an electro-optical (EO) polymer layer, the sensor achieved a high *Q*-factor $>10^{4}$, a sensitivity of 616 nm/RIU, and an *FOM* $>10^{4}\;{RIU}^{-1}$. The strong optical localization at the interface enhanced light–matter interaction, leading to exceptional performance metrics, particularly in narrowband filtering and optical switching. The hybrid resonance system provided robust, precise control over optical properties, making it suitable for high-performance sensing applications in optical communication and integrated photonic circuits. Similarly, Mohamed et al. [66] explored the coupling of the two 1D topological PCs, leveraging Fano resonances. The interference between the TESs of the coupled structures produced sharp resonance peaks with ultrahigh *Q*-factors exceeding $10^{6}$ and sensitivity reaching 888 nm/RIU with a detection limit as low as $10^{-7}$ RIU, demonstrating their potential for detecting minute RI changes (see Figure 1b). The coupling mechanism allowed for the precise manipulation of light propagation, offering significant improvements in sensor performance compared to standalone TES-based systems.

Graded-index topological resonators, such as those proposed by Goyal et al. [96], utilize hyperbolic-graded PCs to create a deliberate modulation of the RI, resulting in tailored dispersion characteristics and the excitation of protected TESs. These structures achieve an impressive sensitivity of 852 nm/RIU, an *FOM* of $>10^{3}\;{RIU}^{-1}$, and a *Q*-factor exceeding 4000 (see Figure 1c). These sensors are ideal for industrial testing, where precise RI changes must be detected under varying conditions.

Several studies have explored integrating graphene-based structures with TPPs to achieve high sensitivity. Hu et al. [97] proposed a high-performance terahertz (THz) RI sensor that combines graphene and multilayer PCs, as shown in Figure 1d. The device achieved a sensitivity of 1.01 THz/RIU and an *FOM* of 631.2 ${RIU}^{-1}$, with tunable performance through chemical potential and incident angle adjustments, making it highly suitable for environmental monitoring in the terahertz spectrum. In another design, Hu et al. [98] integrated graphene-based grating and TPPs to create a multichannel near-infrared sensor, achieving a sensitivity of 950 nm/RIU and an *FOM* of 161 ${RIU}^{-1}$ (see Figure 1e). A dual-channel RI sensor proposed by Huang et al. [64] hybridizes TPPs with defect modes in a 1D PC. This design provides dual-channel functionality. The sensitivities of channel 1 (C1) can reach 120 nm/RIU, and channel 2 (C2) can reach 250 nm/RIU. Enhanced by Fano resonances, the sensor demonstrates improved sensitivity while maintaining robustness against environmental noise, with applications in multiparameter sensing. Similarly, Mathew et al. [65] combined graphene’s tunable plasmonic properties with a 1D topological PC to achieve a high sensitivity of 1.02 THz/RIU and a remarkable *FOM* of 29,410 ${RIU}^{-1}$. The sensor operates in the terahertz region, leveraging localized TESs for RI detection in industrial and biomedical applications.

Further advancements include TES and quasi-BIC-based sensors, leveraging higher order topological modes’ coupling to enhance the robustness against scattering. A high-sensitivity RI sensor using BICs was presented by Zheng et al. [90], achieving high-*Q* resonances ($>10^{4}$) and a *DL* as low as $10^{-5}$ RIU. The merging of topological charges in momentum space enhanced the robustness to scattering and achieved a sensitivity of 36 nm/RIU with an FOM of 5990 ${RIU}^{-1}$ (see Figure 1f). Such designs are particularly suitable for detecting RI changes in fluids and gasses. Sheng et al. [67] proposed an RI sensor based on the Su–Schrieffer–Heeger (SSH) model and the TES-TCS in 2D PCs. The proposed sensor achieved ultrahigh *Q*-factors exceeding $10^{6}$ and a sensitivity of 413.76 nm/RIU in a broad RI range. This device enhanced light–matter interactions by confining light to symmetry-protected corner sites, offering long-term stability and precision. The sensor achieved sub-nanometer resolution for RI variations, which is suitable for trace-level detection.

For advanced RI sensor study based on topological corner modes, O and Kim [99] investigated Fano resonances using topological corner modes based on 2D pseudospin Hall photonic systems. Topological corner states have significantly narrow spectral widths due to their nature as states with single-valued eigenfrequencies independent of the wavenumber, and the resultant Fano resonance has a significantly narrow linewidth and ultrahigh Q-factors in the order of $10^{5}$ and a sensitivity of about 180 nm/RIU with an FOM in the order of $10^{4}$ RIU, enabling applications for index sensing with ultrahigh sensitivity (see Figure 1g). Yu et al. [70] introduced a high-*Q* topological RI sensor using high-order TCSs in Valley Photonic Crystals (VPCs) to achieve ultra-narrow spectral linewidths with a *Q*-factor exceeding $10^{5}$, a high sensitivity of up to 2.06 THz/RIU and a notable FOM standing at $3.86\times 10^{4}$ ${RIU}^{-1}$ (see Figure 1h). The symmetry-protected phases in these states ensured robustness against fabrication imperfections, making the design suitable for THz applications and high-precision material diagnostics. Finally, Abood et al. [100] exploited the unique properties of TCSs and their integration with Fano resonances to achieve high-performance sensing capabilities focused on C4 symmetry. Sharp Fano resonances were created due to the interplay coupling between the TES and TCS. This approach yielded exceptional sensing metrics, including a sensitivity of 439.7 nm/RIU, a *Q*-factor exceeding $10^{6}$, and an *FOM* of over $10^{5}$ ${RIU}^{-1}$ (see Figure 1i). The strong field localization enabled the detection of subtle RI changes. These advancements highlight the transformative potential of TCS-based designs in high-resolution sensing.

RI sensors based on topological photonics leverage advanced physical principles such as TESs, TCSs, Fano resonances, and BICs to achieve unprecedented sensitivity, *Q*-factors, and robust performance. By integrating innovative designs with advanced materials like graphene, these sensors address challenges in conventional photonics and pave the way for cutting-edge applications in healthcare, environmental monitoring, and industrial testing.

### 3.2. Biosensors

Topological photonic biosensors enable the precise detection of specific biomolecules, including cancer biomarkers, SARS-CoV-2 S-glycoproteins, DNA fragments, proteins, viruses, and waterborne bacteria, demonstrating their potential for biomedical diagnostics and disease monitoring with unmatched precision, robustness, and specificity. Their resilience to environmental noise and fabrication imperfections makes them highly suitable for biomedical applications, including disease diagnostics, drug development, and environmental monitoring. The strong light–matter interaction in these biosensors is facilitated by the Berry curvature, which enhances the geometric phase accumulation of light as it interacts with the target biomolecules. This results in sharp resonances and high *Q*-factors, as discussed in Section 2, enabling the precise detection of trace analytes by exploiting unique physical phenomena such as TPPs, topological states, and VPCs. The specificity of these sensors—defined as their ability to selectively detect target analytes in the presence of interfering substances—is enhanced by the sharp resonance peaks and robust edge states, which minimize cross-sensitivity and ensure the accurate detection of target biomolecules [101,102].

A notable example is the polarization-independent topological photonic biosensor proposed by Mingyang Su et al. [69], which utilized 1D topological PCs integrated by adjusting the number of periods. These sensors achieve strong field localization at the interface of metal films and distributed Bragg reflectors, resulting in sensitivities of $1.5677\times 10^{6}\;{RIU}^{-1}$ for the TM mode and $1.3497\times 10^{6}\;{RIU}^{-1}$ for the TE mode (see Figure 2a). Polarization independence is attributed to the inversion symmetry of the photonic bandgap, which ensures the equal sensitivity of TM and TE modes. The sharp resonance peaks and robust edge states of these sensors also contribute to their high specificity, enabling the selective detection of target biomolecules in complex environments. Such devices are particularly effective for applications in food safety, drug diagnostics, and environmental monitoring. In a subsequent study, Mingyang Su et al. [68] extended the TPP biosensor design to enhance the sensitivity and *FOM* further, achieving $2.6553\times 10^{4}\;{RIU}^{-1}$ and an FOM of $3.1238\times 10^{7}\;{RIU}^{-1}{deg}^{-1}$ for the TM polarization mode and $1.3349\times 10^{4}\;{RIU}^{-1}$ and an *FOM* of $6.6745\times 10^{8}\;{RIU}^{-1}{deg}^{-1}$ for the TE polarization mode (see Figure 2b). The high *Q*-factor and robust field confinement make these sensors suitable for detecting trace biomolecules in complex environments.

Furthermore, corner-localized quasi-BICs have enhanced the biosensing performance. Minghao Chao et al. [103] demonstrated a biosensor leveraging higher order topological modes, achieving a sensitivity of 312.8 nm/RIU and an *FOM* exceeding $10^{3}\;{RIU}^{-1}$ (see Figure 2c). The robust light–matter interactions facilitated by the band inversion mechanism enable the precise detection of cancer biomarkers and pathogens, making these sensors indispensable for modern medical diagnostics. The high specificity of these sensors is achieved through their sharp resonance peaks and strong field confinement, which minimize interference from non-target analytes and ensure the accurate detection of target biomolecules [104]. Similarly, Suthar et al. [105] developed a graded thickness defect layer design within a 1D PC to detect waterborne bacteria by tuning the defect modes. This design achieved a sensitivity of 2801.5 nm/RIU and a *Q*-factor of 29,554. The tunable defect modes in this design provide a robust solution for environmental and medical pathogen detection.

**Figure 2 sensors-25-01455-f002:**
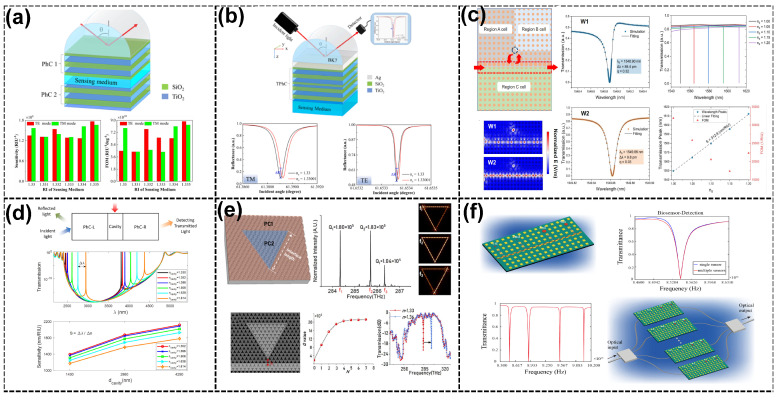
Biosensors. (**a**) Structure of the proposed biosensor with a 1D topological PC with the *S* and FOM of the TE and TM polarization modes [69]. (**b**) Structure of the proposed TPP biosensor with the resonance curves at different values of *n* to show the resonance shift for the TE and TM [68]. (**c**) Schematic of the structure of the defect hybrid waveguide (W2) and CS microcavity with the *E* field in the coupling process and the transmission dips under different values of $n_{0}$ and with the linear fitting for dips in the wavelength with $n_{0}$ and FOMs at different values of $n_{0}$ [103]. (**d**) Schematic of the bio-photonic sensor based on CTCSs with transmission spectra for different concentrations of the SARS-CoV-2 S-glycoprotein solution and the *S* of different optical cavity sizes [106]. (**e**) Schematic diagram of the triangular cavity with the optical spectra of the cavity and the field plots of modes 1, 2, and 3. The SEM of the structure shows the *Q*-values with different values of *N* and the transmission spectra with different values of *n* [59]. (**f**) Schematics of topologically integrated photonic biosensors with wavy lines indicating the topological edge channel and the couplings with the transmission spectrum where multiple photonic biosensors coupled serially to the same edge channel [46].

Another advancement is using conjugated topological cavity states (CTCSs) that provide robust light transport, as proposed by Lin et al. [106]. CTCSs enable the precise detection of biomolecules, such as viral particles. These cavity states maintain high *Q*-factors and robustness against noise, ensuring high sensitivity for biosensing applications, including pandemic monitoring. This mechanism achieved an extreme sensitivity of 2000 nm/RIU and a high *Q* of $10^{9}$ for detecting the concentration of a SARS-CoV-2 S-glycoprotein solution, enabling the detection of viral concentrations with minimal interference (see Figure 2d). In addition, a topological ring resonator (TRS) designed for mid-infrared RI biosensing exhibited a high *Q*-factor of $2.905\times 10^{4}$. Valley–spin locking in topological VPCs ensures robust transmission suitable for early-stage tumor detection [107].

For advances in topological photonic biosensors, Qi Cheng et al. [59] utilized valley edge modes arising from breaking the spatial inversion symmetry. A *Q*-factor of $1.83\times 10^{5}$ was achieved due to the combination of topological protection and low-loss silicon-on-insulator (SOI) platforms and a sensitivity of 1228 nm/RIU, suitable for detecting nanoscale biomarkers such as DNA, proteins, and viruses, with potential applications in disease diagnostics and precision medicine (see Figure 2e). Furthermore, Ze-Lin Kong et al. [46] introduced an integrated biosensing platform where multiple topological photonic biosensors are connected through robust TESs. These integrated systems offer resilience against fabrication imperfections and allow for multiplexing through robust edge–cavity coupling to protect waveguides (see Figure 2f). The network enables the simultaneous detection of multiple biomarkers, making it suitable for high-throughput biomedical diagnostics, such as early cancer screening and monitoring infectious diseases.

In summary, combining topological protection, high-*Q* resonances, and tailored photonic designs offers unparalleled advantages in biomedical sensing. The robustness against noise, environmental perturbations, and fabrication errors ensures reliable and precise detection, while the high specificity of these sensors enables the selective detection of target biomolecules in complex environments. These features position topological photonic biosensors as a cornerstone for the next generation of healthcare diagnostics and environmental monitoring.

### 3.3. Thermal Sensors

Thermal sensors based on topological PCs utilize innovative designs that exploit thermo-optic effects and thermal expansion mechanisms to achieve unparalleled sensitivity and robustness. These systems are crucial for applications in industrial thermal diagnostics. Aly et al. [108] introduced a 1D PC design with defective layers to enhance the sensitivity to temperature variations. The key principle lay in the thermal expansion of materials and the thermo-optic effect, where temperature changes induce shifts in the RI and physical dimensions of the PC layers. The sensor achieved a sensitivity of 0.085 nm/°C and a *Q* of 2216.6, enabling precise temperature monitoring in industrial applications (see Figure 3a). The defect mode shifts observed in these systems are instrumental in detecting minor temperature variations, critical for industrial processes requiring stringent thermal diagnostics. Elsayed et al. [60] improved the hyperbolic metamaterial (HMM) performance within 1D TPCs for temperature sensors. Using silicon–bismuth germinate layers enabled TESs. These TESs facilitated robust light confinement, with the sensor demonstrating a sensitivity of 0.27 nm/°C (see Figure 3b). Integrating HMMs ensures thermal stability and enhances robustness by minimizing the impact of geometrical changes in the photonic structure. This makes such designs ideal for environmental and industrial temperature monitoring applications.

Additionally, Bouazzi et al. [109] proposed a novel approach combining 1D TPCs with nematic liquid crystals (NLCs) to achieve dynamic thermo-optic modulation. By leveraging the RI tunability of NLCs under temperature variations, coupled TES modes exhibited exceptional sensitivity and high-*Q* resonances. Two configurations were explored: the complete replacement of silica layers with NLCs and localized substitution within specific PC regions. These configurations demonstrated sensitivities of up to 0.12317 nm/°C, with a remarkable *FOM* exceeding 1002.48 °C^−1^, suitable for precise temperature sensing in harsh environments (see Figure 3c).

### 3.4. Gas Sensors

Topological photonic gas sensors have been demonstrated to detect nitrogen dioxide (NO_2_), methane (CH_4_), carbon dioxide (CO_2_), and volatile organic compounds (VOCs), making them valuable for environmental monitoring and industrial safety. These sensors leverage the interaction of light with gaseous media, inducing shifts in the resonance or transmission spectra due to RI changes caused by gas adsorption or interaction. Topological PC-based gas sensors’ inherent robustness and high sensitivity are attributed to their unique TES and resonance-enhancing mechanisms. For instance, in the study by Janaszek et al. [110], a 1D photonic hypercrystal integrated with a graphene monolayer achieved extraordinary sensitivity, detecting NO_2_ gas concentrations as low as 1 ppb. The gas molecules altered the carrier density of the graphene layer, inducing shifts in its RI and optical conductivity. This interplay enhanced the TES robustness, ensuring reliable detection even with environmental noise (see Figure 4a). Topological Weyl points in the photonic structure further contributed to the device’s sensitivity by stabilizing the optical states against defects and perturbations, thus ensuring reliable gas sensing in dynamic conditions. In addition, Elshahat et al. [61] also demonstrated that a 1D topological PC heterostructure can enhance the interaction between light and gas molecules due to its robust edge states. In this sensor, analyte gasses replaced the air layers, leading to measurable shifts in the resonance wavelengths due to changes in the effective RI. Two modes were explored: the complete replacement of the air layers with gaseous layers and partial replacement at the interface. These mechanisms achieved a remarkable sensitivity of about 888 nm/RIU and an *FOM* exceeding 10^5^ ${RIU}^{-1}$ (see Figure 4b). The robust TES facilitated these metrics, enabling the precise detection of VOC, CH_4_, and CO_2_ gasses with minimal interference, making it a valuable tool for both environmental monitoring and industrial safety. Further innovations include the work of Fallahi et al. [111], who utilized micro-ring resonators coupled with waveguides to optimize gas detection. The device demonstrated an enhanced sensitivity of 6451 nm/RIU with an *FOM* of 2960 ${RIU}^{-1}$, illustrating the adaptability of TPC designs for diverse sensing applications. By manipulating the coupling region and lattice constant of the micro-ring structures, the sensor achieved exceptional *Q*-factors and sensitivity, applicable to detecting hazardous gasses like CO_2_ and VOCs and even medical diagnostics involving exhaled gasses (see Figure 4c).

### 3.5. Multifunctional and Tunable Sensors

Topological photonic sensors represent a frontier in multifunctional and tunable sensing platforms, integrating diverse functionalities into compact designs. These sensors achieve exceptional performance by combining high-*Q* resonances, multichannel capabilities, and advanced materials, making them indispensable for optical communication, environmental monitoring, and industrial diagnostics. This section delves into key advancements in this domain, providing detailed insights into the underlying physics and engineering. One prominent example is the development of multifunctional PCs with C4 symmetry, which integrate trivial and nontrivial unit cells to create multiple output ports and multiband edge states for applications in optical communication. The interplay of trivial and nontrivial topological modes allows for seamless frequency routing while maintaining robustness. Moreover, the devices demonstrated a sensitivity of as high as 721 nm/RIU, alongside robustness against structural imperfections, making them suitable for real-time environmental monitoring and telecommunication systems [112] (see Figure 5a). Further advancements include heterostructure-based sensors that combine two distinct 1D PC systems to optimize dispersion characteristics. This design leverages the differential bandgaps to create highly sensitive interfaces for the simultaneous detection of RI, temperature, and pressure changes [63]. The multichannel functionality of these sensors reduces device footprints while enhancing data processing capabilities, making them ideal for optical networks and industrial process monitoring (see Figure 5b).

The integration of graphene with topological PCs has also led to significant breakthroughs [113]. These systems exploit the interaction between graphene’s plasmonic modes and TESs, resulting in tunable Fano resonances. These devices achieve reconfigurability by applying an external bias to graphene, offering the precise detection of RI changes and signal switching, vital for dynamic telecommunication networks and real-time environmental sensing (see Figure 5c). Another fascinating advancement involves coupling defect modes with the TES, as shown by [117], producing sharp Fano resonances. The tunability of these resonances through external magnetic fields enhances sensor sensitivity and operational flexibility. For instance, coupling micro-ring resonators with PC nanobeams in silicon nitride platforms allows for dual-band operation and high sensitivity in telecom bands. Such designs are instrumental for optical signal processing and dual-parameter sensing. Moreover, Mendoza-Castro et al. [118] proposed a coupling of Fano resonances with micro-ring resonators in silicon nitride PCs, enabling dual-band operation. The enhanced Fano resonances, created by coupling micro-ring resonators with PC nanobeams, enable high sensitivity and versatile operation in dual telecom bands. The dual-band operation and high sensitivity of specific topological photonic sensors make them suitable for optical signal processing. This design shows significant versatility in detecting physical and biochemical parameters in telecommunication and sensing applications.

The use of non-Hermitian multilayered heterostructures has further pushed the boundaries of sensor technology. By combining Fano resonances with exceptional points (EPs), these systems achieve ultrahigh *Q*-factors exceeding 3500 and sensitivity to RI changes, stress, and temperature fluctuations. This makes them ideal for advanced optical diagnostics and high-precision environmental monitoring [114] (see Figure 5d). The work by K. Huo et al. [115] integrated Weyl semimetals (WSMs) into 1D PCs to create tunable Fano resonances. WSMs exhibit unique bulk and surface states interacting with photonic modes to produce sharp and tunable resonances. By adjusting the Fermi energy of the Weyl semimetal through external fields, the spectral position and linewidth of the Fano resonance can be dynamically controlled. This enables real-time tunability, which is crucial for multiparameter sensing (see Figure 5e). The sensor demonstrates high sensitivity for RI changes and can switch optical signals, making it valuable for telecommunication and sensing applications. Lastly, Abood et al. [116] introduced hybrid PCs leveraging topological multi-Fano resonances confined to quasi-BICs. These systems exhibited ultrahigh *Q*-factors of $10^{8}$ and a remarkable *FOM* of above $10^{7}$ ${RIU}^{-1}$ (see Figure 5f). The design achieved a tunability of 320–392 nm/RIU, with a minimal device size, opening possibilities for on-chip integration, making it an ideal platform for on-chip, ultra-compact sensors.

## 4. Performance Metrics and Comparative Insights in Topological Photonic Sensors

Topological photonic sensors derive their exceptional performance from a combination of physical mechanisms and advanced materials. The sensing principle based on TESs, TCSs, Fano resonances, quasi-BICs, TPPs, and VPCs offers unique advantages regarding robustness, sensitivity, and environmental stability. A key metric for evaluating their performance is sensitivity, which quantifies the change in an optical property due to variations in the environmental parameters. Additionally, the *Q*-factor and *FOM* provide insights into the energy efficiency and resolution of the sensors. For instance, TESs enable unidirectional light propagation that is immune to backscattering along the edges. This property is pivotal for sensors operating in noisy environments or under fabrication imperfections. Meanwhile, TCSs leverage high-symmetry points to confine light, resulting in extreme localization and sensitivity to minute changes.

### 4.1. Fano Resonances and Their Tunability

Fano resonances occur due to the interference between discrete localized states (e.g., edge or corner states, or even defect modes) and continuum states [3,119,120,121,122,123]. This interference creates sharp asymmetric spectral profiles with enhanced sensitivity to changes in the RI. Coupling Fano resonances with topological protection combines their spectral sharpness with robustness to scattering, resulting in sensors with ultrahigh sensitivity and low detection limits [3,122,124,125]. This kind of asymmetric resonance profile allows for high-resolution sensing and is suitable for multiparameter sensing in complex environments. The integration of graphene with TPCs further enhances this phenomenon by introducing plasmonic modes that strongly interact with photonic edge states. The optical properties of graphene, such as its RI and carrier density, can be dynamically tuned using an external bias, enabling real-time reconfigurability. The interaction between graphene’s plasmonic modes and TESs produces tunable Fano resonances. In dynamic signal routing, tunable Fano resonances enable precise optical switching for telecommunication due to their sharp resonance profiles [3,119,121,122,126,127,128,129]. Moreover, they allow for the detection of trace biomarkers, making them suitable for early disease diagnostics in biosensing.

### 4.2. Quasi-Bound States in the Continuum (Quasi-BICs)

Quasi-BICs are another type of localized mode where destructive interference traps light in a resonant state while maintaining coupling to a continuum of radiation modes [130,131,132]. These states are characterized by sharp spectral features and ultrahigh *Q*-factors, achieved through fine-tuning the geometric or material parameters. Corner states provide extreme light confinement, enhancing the sensitivity of sensors by increasing the overlap between the optical field and the target analytes [133]. Quasi-BICs achieve ultrahigh *Q*-factors due to the suppression of radiative losses, enabling the precise detection of minute molecular changes [134]. Both mechanisms offer robustness to defects and environmental disturbances, ensuring reliable sensing performance.

When quasi-BICs are combined with multiple Fano resonances, these systems achieve ultrahigh *Q*-factors and an exceptional *FOM* [135,136]. The coupling of multiple resonances allows for the simultaneous detection of multiple parameters within a single compact device. The compact design of these systems makes them ideal for on-chip integration, addressing the demand for ultra-compact, high-performance sensing platforms. Multi-Fano resonances leverage topological states to achieve ultrahigh *Q*-factors and exceptional spectral control. Using quasi-bound states in the continuum ensures minimal scattering and maximum light–matter interaction.

Corner states and quasi-BICs have been utilized in biological sensing to detect cancer biomarkers and pathogens. For instance, higher order modes facilitate robust light–matter interactions, achieving exceptional sensitivity and FOMs. The symmetrical protection in these systems ensures consistent performance even in noisy or imperfect environments. Also, this approach is efficient for fluid-based sensing, where RI variations are subtle but critical.

### 4.3. Weyl Semimetals and Topological Light–Matter Interaction

Weyl semimetals (WSMs) are quantum materials with unique electronic band structures featuring Weyl points—singularities where conduction and valence bands meet [137]. These materials interact with photonic modes to produce tunable Fano resonances when integrated into photonic systems. By dynamically adjusting the Fermi energy of the WSM via external fields, these resonances’ spectral position and linewidth can be finely controlled. WSMs integrated with topological PCs can be demonstrated to achieve tunability for the real-time detection of RI changes [138,139]. These sensors are especially valuable for multiparameter sensing applications, such as simultaneous temperature, pressure, and RI detection. Tunable WSM-based sensors ensure precise signal switching and wavelength multiplexing. The real-time sensing of the temperature and stress enhances safety and process control.

### 4.4. Conjugated Topological Cavity State (CTCS)

A CTCS is formed by coupling multiple topological cavities within a photonic system [106,140]. These cavity states combine the robustness of topological protection with the enhanced light–matter interaction provided by optical cavities. The coupling between the cavities is mediated by topologically protected waveguides, which ensure that the optical signal remains immune to backscattering or scattering losses. CTCSs exhibit ultrahigh *Q*-factors and exceptional sensitivity due to cavity enhancement and topological robustness. The light within these cavities interacts strongly with the surrounding medium, making CTCS-based sensors highly effective for detecting biomolecules, such as viral particles or biomarkers of cancer. Ultrahigh *Q*-factors enable a sharp spectral resolution, which is critical for detecting subtle changes in molecular concentrations. The topological protection ensures reliable performance under noisy and imperfect conditions, such as in pandemic monitoring. The strong field localization in CTCSs enhances light–matter interaction, improving the sensitivity for trace analyte detection. CTCSs have been used to detect viral particles like SARS-CoV-2 with extreme sensitivity. By monitoring the spectral shifts caused by molecular binding, these sensors achieve detection limits at the nanoscale, making them indispensable for virology and pandemic monitoring applications.

### 4.5. Tamm Plasmon Polaritons (TPPs)

TPPs are localized electromagnetic states formed at the interface of a metallic layer and a distributed Bragg reflector (DBR) [141,142]. The physical origin of TPPs lies in the phase-matching conditions between the evanescent fields of the metallic surface plasmon and the photonic bandgap of the DBR [143,144,145]. When these conditions are satisfied, TPPs emerge as highly localized modes with strong field confinement at the interface. TPPs exhibit unique polarization independence arising from the inversion symmetry of the photonic bandgap. Unlike traditional plasmonic systems, where TE and TM modes exhibit different sensitivities, TPPs maintain identical responses for both polarizations, making them ideal for biosensors that require consistent performance across different operational conditions. The high field confinement at the TPP interface enhances the interaction between light and the surrounding medium, leading to improved sensitivity to RI changes. Polarization independence ensures versatility in sensing, allowing for the detection of various analytes with consistent accuracy. The strong coupling of TPPs to the photonic structure ensures robustness against fabrication defects and environmental noise, which is critical for real-world biomedical applications. In biosensing, TPPs detect biological analytes by monitoring shifts in the resonant wavelength or intensity caused by the binding of biomolecules to the sensor surface. The robustness and high field intensity of TPP-based sensors make them highly suitable for applications in food safety, drug diagnostics, and environmental pathogen detection.

### 4.6. Valley Photonic Crystals (VPCs)

VPCs exploit valley degrees of freedom in photonic band structures to create robust, low-loss edge states [146,147,148]. Valleys are local extrema in the band structure that arise from breaking the spatial inversion symmetry. The associated Berry curvature in these valleys results in robust optical transport along valley edges, analogous to electronic transport in valley electronic systems [149,150]. Valley edge states are topologically protected and exhibit unidirectional propagation at the interface between two regions with opposite valley Chern numbers [151,152,153]. This protection ensures that the edge states remain robust against backscattering, even in the presence of defects or structural imperfections. Valley edge modes offer high field confinement, which enhances the interaction between light and biomolecules on the sensor surface. The topological protection of valley modes ensures low-loss light propagation, improving the accuracy and reliability of biosensors. VPCs are compatible with SOI platforms, making them suitable for large-scale integration and mass production. In biosensors, VPCs detect nanoscale biomarkers such as DNA, proteins, and viruses. The strong confinement of valley edge modes enables precise measurements of RI changes caused by biomolecular binding events, with applications in disease diagnostics and precision medicine.

### 4.7. Exceptional Points (EPs)

Exceptional points (EPs) occur in non-Hermitian systems where two or more eigenvalues and their corresponding eigenvectors coalesce [154,155]. At these singularities, the system becomes highly sensitive to external perturbations, enabling the detection of minute changes in the environmental or physical conditions. The coupling of Fano resonances with EPs in multilayered heterostructures leads to enhanced spectral shifts and ultrahigh *Q*-factors [156,157]. These systems leverage the intrinsic instability near EPs to magnify subtle RI or temperature variations. Optical diagnostics: The high sensitivity of EP-enhanced sensors is ideal for detecting small-scale material changes, such as in the stress and strain in optical components. EPs allow for the real-time detection of temperature fluctuations or pollutant concentrations in dynamic environments.

## 5. Challenges and Future Directions for Topological Photonic Sensors

Topological photonic sensors face several challenges, including material limitations, scalability, environmental stability, and real-world implementation. These obstacles can be overcome by applying advanced physical principles, designing novel structures, integrating cutting-edge materials, and addressing the need for multifunctional, dynamic platforms. Here, we delve deeper into how these challenges can be addressed, focusing on physics-based solutions and the mechanisms behind them.

### 5.1. Material Challenges and Scalability

Many topological photonic designs rely on advanced materials such as graphene, WSMs, and HMMs. These materials exhibit exceptional properties but face challenges such as high fabrication costs, limited scalability, and compatibility with conventional CMOS technologies [24,158,159,160]. By integrating advanced materials with SOI platforms, the scalability and cost-effectiveness of silicon photonics can be combined with the exceptional performance of topological materials. For example, graphene’s plasmonic properties can enhance light–matter interaction when deposited on silicon photonic waveguides [161,162]. The strong optical field confinement in SOI platforms increases the interaction cross-section, improving sensitivity. The substantial field enhancement near the graphene–SOI interface results from the coupling between silicon’s high RI and graphene’s tunable plasmonic resonance. This coupling can be dynamically tuned by altering the Fermi energy of graphene through an external bias, enabling cost-efficient, scalable, and reconfigurable sensors.

For Weyl semimetals, leveraging epitaxial growth techniques can produce large-scale thin films compatible with photonic devices [163,164]. This maintains their topological protection while enhancing light–matter interaction through the strong Berry curvature near Weyl points.

By integrating TIs with microfluidic platforms, THz biosensors can detect trace biomolecules such as cancer biomarkers, proteins, and DNA. THz waves interact with the vibrational and rotational modes of biomolecules, making them ideal for label-free sensing. The robustness of TI waveguides ensures high sensitivity even when the biosensor is subjected to environmental fluctuations. Also, SOI-based VPCs support valley-protected edge states that enhance light–matter interaction by slowing down light propagation. This allows for a longer interaction time between light and biomolecules, improving the sensitivity to low-concentration analytes such as viruses, antigens, and biomarkers. These sensors are particularly suited for the real-time monitoring of biological samples in point-of-care diagnostics.

### 5.2. Environmental Stability and Noise Interference

While topological photonic sensors are inherently robust against defects and scattering, external factors such as temperature fluctuations, humidity, or vibrations can still degrade their performance. Developing thermally stable designs, such as those employing hyperbolic metamaterials or liquid crystals, can mitigate these issues. HMMs, composed of alternating metal–dielectric layers, can provide thermal stability by suppressing temperature-induced changes in the RI. The highly anisotropic optical properties of HMMs confine light in sub-wavelength volumes, reducing environmental interference. The thermal–optic coefficient of HMMs can be tailored by choosing materials with complementary temperature-dependent refractive indices. For example, combining a material with a positive dn/dT (e.g., silicon) with a material with a negative dn/dT (e.g., certain dielectrics) can create a composite structure with near-zero thermal sensitivity. TESs confined within photonic bandgaps are mainly immune to scattering and fabrication imperfections. By incorporating temperature-insensitive materials into the PC lattice, the bulk–edge correspondence principle ensures that edge states remain stable under environmental fluctuations.

The robustness of topological photonic sensors against noise is further caused by their high SNR. The topological protection of edge states ensures that light propagates along the material boundaries with minimal scattering, even in the presence of defects or environmental noise. This results in a strong, well-defined signal (e.g., resonance shift) relative to the background noise, leading to a high SNR [165,166]. For example, in RI sensors, the sharp resonance peaks associated with Fano resonances or quasi-bound states in the continuum (quasi-BICs) enable the precise detection of small RI changes, even in noisy environments. The high SNR of these sensors makes them particularly suitable for applications in industrial settings or complex biological samples, where environmental noise is often significant.

### 5.3. Dynamic Tunability and Reconfigurability

Many existing sensors are designed for fixed operational parameters, limiting their ability to adapt to dynamic environments or multifunctional sensing tasks. Integrating optically or electrically tunable materials like WSMs, graphene, or liquid crystals can enable real-time reconfigurability [167]. For example, liquid crystals exhibit anisotropic refractive indices that can be electrically or thermally tuned to modulate the propagation characteristics of edge states. The RI of liquid crystals is susceptible to electric fields due to their molecular alignment properties. Combined with topological photonic structures, this tunability allows for precise control over the resonance frequencies. Similarly, graphene’s carrier density can be modulated via an external bias, shifting the spectral position of plasmonic modes and enabling real-time tunability.

Non-Hermitian photonic systems featuring EPs provide extreme sensitivity to perturbations. By coupling Fano resonances with EPs, small environmental changes (e.g., RI or temperature variations) can induce significant spectral shifts, enabling high-resolution dynamic sensing. At EPs, the eigenvalues and eigenvectors of the system coalesce, leading to a square root dependence on perturbations. This nonlinear response amplifies subtle environmental changes, making the system highly sensitive to dynamic variations.

Using optically or thermally tunable substrates such as high-resistivity silicon (HR-Si), these PTIs can dynamically modulate the photonic bandgap to detect subtle changes in the RI caused by molecular binding. This capability is significant for multifunctional sensors, where reconfigurability is essential for detecting a wide range of analytes. The robustness of these devices makes them suitable for deployment in harsh environments or remote monitoring stations. In addition, thermal sensors govern light–matter interactions under temperature variations. These mechanisms include the thermo-optic effect, thermal expansion, and enhanced light confinement in defective modes.

### 5.4. Miniaturization and On-Chip Integration

Achieving ultra-compact designs with multifunctional capabilities requires balancing high-*Q* resonances, robust sensing, and on-chip compatibility. Higher order topological modes, such as TCSs confined to geometric corners, enable extreme light confinement and compact device designs. These modes leverage symmetry-protected mechanisms (e.g., C4 or C6 lattice symmetries) to ensure robust performance even in miniaturized structures. The localization of light at high-symmetry points reduces radiative losses and enhances the *Q*-factor. The rotational symmetry of the lattice ensures robustness against structural imperfections, making these sensors ideal for compact on-chip integration. Quasi-BICs provide ultrahigh *Q*-factors by suppressing radiative losses near symmetry-protected points in the photonic band structure. These systems can be engineered to support multiple sensing functionalities (e.g., detecting the temperature, pressure, and RI) within a single, compact device. The high *Q*-factor arises from destructive interference between radiative modes, which traps light within the structure for extended durations. This extended interaction time increases the sensitivity and resolution of the sensor.

Developing sensors that simultaneously detect multiple parameters (e.g., temperature, pressure, and chemical composition) while remaining compact and efficient is a significant challenge. Combining Fano resonances, quasi-BICs, and EPs within a single device makes it possible to sense multiple parameters simultaneously. For example, Fano resonances can detect RI changes, while EPs amplify the temperature sensitivity. The coexistence of multiple resonances creates a rich spectral landscape where each resonance is sensitive to a specific parameter. By carefully tuning the material and geometric properties of the photonic structure, these resonances can be independently optimized for multifunctional sensing.

Finally, exploring novel materials, such as 2D transition metal dichalcogenides (TMDs) or hybrid organic–inorganic perovskites, could provide new avenues for enhancing light–matter interaction and tunability in topological photonic sensors. Machine learning algorithms could optimize photonic structures for specific sensing tasks, identifying configurations with the maximum sensitivity and *Q*-factors while minimizing fabrication challenges. Integration with Internet of Things (IoT) platforms: Embedding topological photonic sensors into IoT devices could enable the real-time, remote monitoring of environmental, biomedical, and industrial parameters.

By addressing these challenges with innovative physics-based solutions, topological photonic sensors can fully realize their potential as next-generation tools for precision sensing across diverse scientific and industrial domains.

### 5.5. Practical Feasibility of Topological Photonic Sensors

The practical feasibility of topological photonic sensors varies significantly depending on the sensor type and application. Below, we discuss the most promising directions and their practical implementation challenges.

RI sensors and thermal sensors are among the most practically feasible topological photonic sensors. RI sensors, such as those based on TESs and Fano resonances, have been experimentally demonstrated to have high sensitivity and robustness against noise. For example, the 1D topological photonic crystal heterostructure proposed by Gao, Hu et al. [168] experimentally reproduces Fano resonance in the optical communication range with a high quality of 10^4^ and has been tested in real-world environments. Similarly, thermal sensors leveraging the thermo-optic effect and thermal expansion, such as those proposed by Aly et al. [108], have shown high precision and stability, making them suitable for industrial and environmental monitoring.

Biosensors and gas sensors face more significant challenges in practical implementation. While biosensors based on TPPs and CTCSs have demonstrated high sensitivity in detecting biomolecules, their practical application is hindered by the need for functionalized surfaces and the complexity of integrating these sensors into portable devices. Gas sensors face selectivity and environmental stability challenges, particularly those designed to detect trace gasses like NO_2_ and CO_2_. For instance, the graphene-based gas sensor proposed by Janaszek et al. [110] achieved a detection limit of 1 ppb for NO_2_ but requires further optimization to minimize cross-sensitivity to other gasses and improve the long-term stability.

Some emerging sensor technologies, such as non-Hermitian sensors based on EPs and Weyl semimetal-based sensors, remain primarily theoretical. While EP-based sensors can achieve ultrahigh sensitivity due to eigenvalue coalescence, their fragile nature and high sensitivity to perturbations make them challenging to implement in practical devices. Similarly, Weyl semimetal sensors, which rely on topological surface states and strong Berry curvature effects, require low-temperature operation, making them impractical for everyday sensing applications. Although these approaches hold promises for future high-precision sensors, they have not yet reached the level of experimental validation necessary for real-world deployment.

## 6. Conclusions

Topological photonic sensors represent a significant leap in sensing technologies. The theoretical principles of topological photonics, such as the Chern number and Berry curvature, provide the foundation for the robust and sensitive performance of topological photonic sensors. These principles enable the creation of defect-immune edge states, enhanced light–matter interactions, and sharp resonances, which are experimentally exploited in refractive index sensors, biosensors, and gas sensors. By leveraging these topological properties, next-generation sensors can achieve unprecedented sensitivity and reliability in diverse applications. By harnessing mechanisms utilizing TESs, TCSs, quasi-BICs, TPPs, and VPCs, these sensors exhibit unparalleled sensitivity and robustness against fabrication defects and environmental noise. RI sensors leverage TESs and TCSs to detect minute changes in the surrounding medium, while biosensors integrate TPPs and VPCs to detect biological analytes and pathogens precisely. Gas sensors, operating on resonance shifts driven by RI variations, provide exceptional sensitivity for trace gas detection, and thermal sensors exploit thermo-optic effects and thermal expansion to monitor temperature variations with high precision.

The robust light–matter interaction facilitated by topological protection makes these sensors ideal for next-generation applications ranging from chemical detection and biomedical diagnostics to quantum metrology and industrial monitoring. However, despite these advancements, challenges remain, particularly in fabricating large-scale, defect-free photonic structures and achieving stable tunability in dynamic sensing environments. Advancements in hybrid materials such as graphene, Weyl semimetals, and hyperbolic metamaterials hold the potential to address these limitations by offering tunability, enhanced field confinement, and compatibility with integrated photonic circuits.

Future research should focus on refining the physical design of topological PCs, mainly through the controlled manipulation of topological phase transitions and coupling phenomena like Fano resonances and EPs. These developments will unlock new functionalities in real-time multiparameter and tunable sensing platforms, expanding the applicability of topological photonic sensors in fields such as telecommunications, precision medicine, and environmental protection.

## Figures and Tables

**Figure 1 sensors-25-01455-f001:**
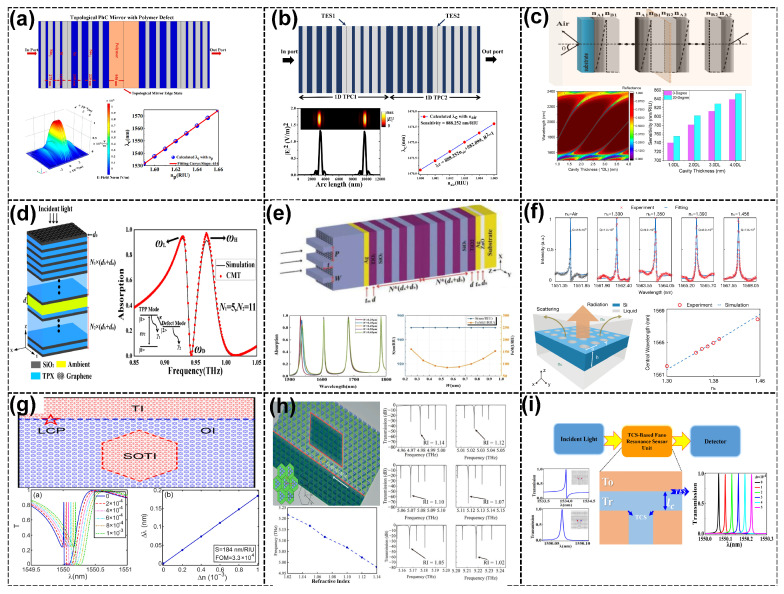
RI sensors. (**a**) Schematic diagram of the 1D topological PC mirror heterostructure with an EO polymer defect layer with the *E* field distribution at λ=1514.58 nm and $\lambda_{c}$ vs. $n_{p}$ variation [95]. (**b**) Diagram of the coupling of two 1D topological PCs with the *E* field distribution at the two interface states and $\lambda_{c}$ vs. RI variation in air [66]. (**c**) Hyperbolic graded topological PC with cavity thickness-dependent resonance λ variations and the comparative sensitivity analysis of four different cavity thicknesses at 0- and 20-degree incidence angles [96]. (**d**) Hybrid structure and absorption spectra of the sensor for various values of $n_{A}$ with the effect of the incident angle *θ* on S and the FoM [97]. (**e**) Diagram of a hybrid Tamm structure with the graphene-based grating’s effect on the sensitivity and quality factor of the GMR peak of the sensor [98]. (**f**) Measured spectra of high-*Q* resonances for different values of $n_{c}$. The intensities characterized by a photodetector (markers) are fitted with Fano line shapes (lines) as a function of λ with a PC schematic for RI sensing. $\lambda_{c}$ shift as $n_{c}$ is tuned from 1.300 to 1.456 [90]. (**g**) Photonic structure of pseudospin Hall topological edge waveguide coupled to the cavity supporting the corner states with transmission Fano resonance spectra depending on the index change Δ*n* of the surrounding medium [99]. (**h**) Diagram of the topological RI sensor. Variations in the transmission rate in response to changes in the background RI with the relationship between the corner state’s $\lambda_{c}$ and the RI [70]. (**i**) Schematics of the coupling structure between the TES and TCS with the transmission spectrum as a function of $\lambda_{c}$ due to a changing RI and possibly different topological Fano resonance peaks [100].

**Figure 3 sensors-25-01455-f003:**
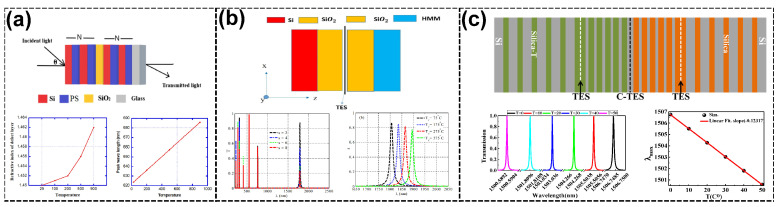
Thermal sensors. (**a**) Structure of the proposed 1D defected PC, showing the change in the RI of the defect layer and the peak wavelength with the change in temperature at a normal incidence [108]. (**b**) Diagrammatic representations of the 1D TPC structures. One-dimensional TPC transmittance characteristics included the various periodicities of the HMM and transmittance at different temperature values [60]. (**c**) Diagram of the NLC configurations and the blueshift of the resonance wavelength with an increasing temperature from *T* = 0 to *T* = 50 °C, showing the resonance shifting as a function of *T* [109].

**Figure 4 sensors-25-01455-f004:**
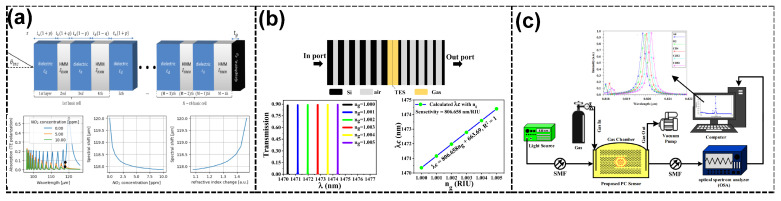
Gas sensors. (**a**) Schematical of the 1D PC hypercrystal revealing the Weyl point and spectral shift in the peak absorption as a function of the wavelength, as well as the NO_2_ concentration and change in graphene’s RI for the sensor, revealing Type II dispersion [110]. (**b**) Diagram of the 1D topological PC with replacement interface layers with different gaseous samples and the transmission spectrum with varying values of $n_{g}$ and the resonance shift as a function of the RI of the gas sample [61]. (**c**) Proposed experimental setup of the proposed gas sensor and the position of the resonance peak as a function of the RI for air, O_2_, CH_4_, C_2_H_2_, and C_2_H_6_ sensor devices [111].

**Figure 5 sensors-25-01455-f005:**
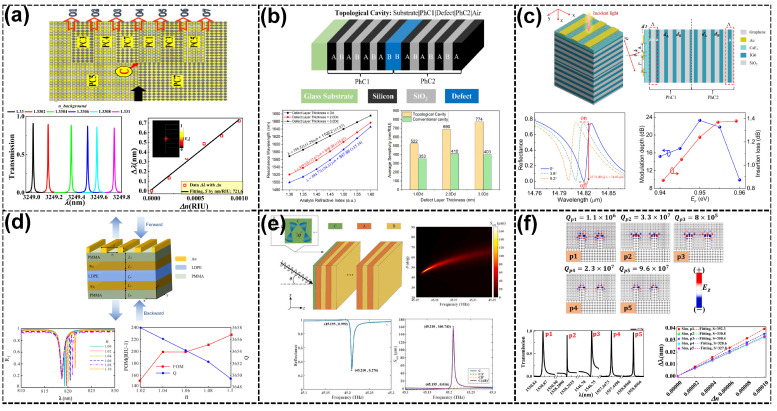
Multifunctional and tunable sensors. (**a**) The structure supports seven distinct output ports from O1 to O7 with a central cavity, C, at the port’s deviation. The transmission spectrum is a function of the cavity resonance due to the change in the RI of the surrounding medium, and the resonance shifts ∆λ as a function of ∆n [112]. (**b**) Topological cavity structure and resonance shifting as a function of *n* at different defect layer thicknesses, Dd, 2Dd, and 3Dd [63]. (**c**) Hybrid structure for the GSPP–TES coupling of the top and front views. The reflection spectrum with the incident angles θ = 0°, 3.8°, and 5.2° [113]. (**d**) Diagram of the non-Hermitian optical heterostructure, with forward reflection spectra with different ambient n values with the FOM and *Q* value of the obtained peaks [114]. (**e**) Schematic of the heterostructure; *Q* indicates the Weyl node separation of the WSM and the GH shift as functions of the frequency and incident angle under TM polarization. Narrow Fano resonance in the heterostructure C (AB)7 and GH shifts as functions of the incident angle for different structures [115]. (**f**) Multi-Fano resonances in the transmission spectrum with unique on/off switching patterns of highly localized electric field modes at the four corners. The resonance wavelength shifts Δλ as a function of Δ*n* for the five multi-Fano resonances [116].

## Data Availability

The data underlying the results presented in this paper are available on request.

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
