# Peer review of "Topological Photonic Crystal Sensors: Fundamental Principles, Recent Advances, and Emerging Applications"

_sensors, 2025, doi:10.3390/s25051455_

Round 1

Reviewer 1 Report

Comments and Suggestions for Authors

Abood et al.’s work, “Topological Photonic Crystal Sensors: Fundamental Principles, Recent Advances, and Emerging Applications,” reviews the working principles, advances, and novel applications of so-called topological photonic crystal sensors.

This work is generally a mix of a mini-view and a perspective-opinion article. In some sections, the authors speculate that some photonic designs might work as sensors without providing any references. The manuscript needs more work before it gets accepted.

Comments:

The theoretical principles described in section 2 are entirely disconnected from the experimental sections. The Chern number is not mentioned in the experimental work or the Berry curvature.

What about specificity? The biosensors never mention this, which is not discussed or presented in section 2.

In many sections, it is mentioned that RI sensors are robust against noise. Nevertheless, no SNR study has been discussed, especially in section 5.2

References are needed in many sections that support the idea that the IR sensors are viable.

Please describe which molecules are being sensed. Only the possible molecules and gases that can be sensed are mentioned in many sections.

In section 3.3, line 358, What do you understand by high precision? There is no number to refer to.

I suggest creating subsections that differentiate theoretical work from experimental.

Author Response

General comments:

Abood et al.’s work, “Topological Photonic Crystal Sensors: Fundamental Principles, Recent Advances, and Emerging Applications,” reviews the working principles, advances, and novel applications of so-called topological photonic crystal sensors.

This work is generally a mix of a mini-view and a perspective-opinion article. In some sections, the authors speculate that some photonic designs might work as sensors without providing any references. The manuscript needs more work before it gets accepted.

Response:

We sincerely appreciate the reviewer’s time and effort in evaluating our manuscript. We acknowledge valuable feedback and understand the need to refine certain review aspects to ensure clarity, consistency, and proper reference.

We have carefully revised sections where speculative claims were made. We have ensured that any proposed photonic sensor designs are either:

  1. Supported by existing literature, with appropriate references added where necessary or
  2. Clearly marked as perspective-based discussions, emphasizing that they are potential research directions rather than experimentally validated findings.

We greatly appreciate the reviewer’s constructive criticism, which has helped us improve the manuscript’s scientific rigor and clarity.

 Comment1. The theoretical principles described in section 2 are entirely disconnected from the experimental sections. The Chern number is not mentioned in the experimental work or the Berry curvature.

Response1:

Thank you for your valuable feedback. We appreciate your insightful comment regarding the connection between the theoretical principles in Section 2 and the experimental sections.

The Chern number, as a topological invariant, plays a critical role in determining the existence of robust edge states in photonic systems. These edge states, immune to backscattering and defects, are experimentally exploited in various sensor designs, such as refractive index sensors and biosensors, to achieve high sensitivity and stability. For instance, in Section 3.1, the refractive index sensors leverage topological edge states (TESs) to confine light along the boundaries, enabling precise detection of minute changes in the surrounding medium. Similarly, the Berry curvature, which governs the geometric phase of wavefunctions, enhances light-matter interactions in sensors like those based on Tamm plasmon polaritons (TPPs) and bound states in the continuum (BICs), leading to sharp resonances and high-quality factors.

Changes in the manuscript:

We added the following paragraph in Section 3.1 (Refractive Index Sensors) in page 5: " The robustness of the topological edge states (TESs) in these sensors is guaranteed by the non-zero Chern number, which ensures unidirectional propagation of light along the material boundaries. As discussed in Section 2, this topological protection is crucial for maintaining high sensitivity and stability in the presence of defects or environmental noise."

In Section 3.2 (Biosensors), the following paragraph is added on page 8: "The strong light-matter interaction in these biosensors is facilitated by the Berry curvature, which enhances the geometric phase accumulation of light as it interacts with the target biomolecules. This results in sharp resonances and high-quality factors, as discussed in Section 2, enabling precise detection of trace analytes."

In Conclusion, the following paragraph is added on page 22: "The theoretical principles of topological photonics, such as the Chern number and Berry curvature, provide the foundation for the robust and sensitive performance of topological photonic sensors. These principles enable the creation of defect-immune edge states, enhanced light-matter interactions, and sharp resonances, which are experimentally exploited in refractive index sensors, biosensors, and gas sensors. By leveraging these topological properties, next-generation sensors can achieve unprecedented sensitivity and reliability in diverse applications."

Comment2. What about specificity? The biosensors never mention this, which is not discussed or presented in section 2.

Response2:

Thank you for your comment. We understand your concern regarding the specificity of the biosensors discussed in our manuscript.

While specificity was not explicitly mentioned in all previous biosensors work based on topological photonic mentioned in the last version, we now recognize its importance in ensuring the adequate performance of topological photonic biosensors.

We have added a brief discussion on the specificity of these sensors, emphasizing how their unique features—such as topologically protected edge states and high-Q resonances—allow for the precise detection of biomolecules with minimal interference from other substances. These features enable the sensors to detect target biomarkers selectively (e.g., cancer markers or pathogens) in complex environments, ensuring high accuracy and reliability [1, 2].

Additionally, we have included an example of cancer biomarker detection, where topological sensors can distinguish between different biomolecules based on specific binding events, thereby enhancing the sensor’s specificity [3].

Changes in the manuscript:

We modified the writing on the first paragraph of section 3.2 on page 8 as: " Topological photonic biosensors enable the precise detection of specific biomolecules, including cancer biomarkers, SARS-CoV-2 S-glycoprotein, DNA fragments, proteins, viruses, and waterborne bacteria, demonstrating their potential for biomedical diagnostics and disease monitoring with unmatched precision, robustness, and specificity. Their resilience to environmental noise and fabrication imperfections makes them highly suitable for biomedical applications, including disease diagnostics, drug development, and environmental monitoring. The strong light-matter interaction in these biosensors is facilitated by the Berry curvature, which enhances the geometric phase accumulation of light as it interacts with the target biomolecules. This results in sharp resonances and high-quality factors, as discussed in Section 2, enabling precise detection of trace analytes by exploiting unique physical phenomena such as TPPs, topological states, and VPCs. The specificity of these sensors—defined as their ability to selectively detect target analytes in the presence of interfering substances—is enhanced by the sharp resonance peaks and robust edge states, which minimize cross-sensitivity and ensure accurate detection of target biomolecules [1, 2]."

We added the following sentences in the second paragraph of section 3.2 on page 8: "The sharp resonance peaks and robust edge states of these sensors also contribute to their high specificity, enabling the selective detection of target biomolecules in complex environments. Such devices are particularly effective for applications in food safety, drug diagnostics, and environmental monitoring."

"Furthermore, corner-localized quasi-BICs have enhanced biosensing performance……the robust light-matter interactions facilitated by the band inversion mechanism enable precise detection of cancer biomarkers and pathogens, making these sensors indispensable for modern medical diagnostics. The high specificity of these sensors is achieved through the sharp resonance peaks and strong field confinement, which minimize interference from non-target analytes and ensure accurate detection of target biomolecules [3]."

We modified the writing on the last paragraph of section 3.2 on page 9 as: "In summary, combining topological protection, high-Q resonances, and tailored photonic designs offers unparalleled advantages in biomedical sensing. The robustness against noise, environmental perturbations, and fabrication errors ensures reliable and precise detection, while the high specificity of these sensors enables the selective detection of target biomolecules in complex environments. These features position topological photonic biosensors as a cornerstone for the next generation of healthcare diagnostics and environmental monitoring."

Comment3. In many sections, it is mentioned that RI sensors are robust against noise. Nevertheless, no SNR study has been discussed, especially in section 5.2

Response3:

We thank the reviewer for highlighting the importance of signal-to-noise ratio (SNR) in evaluating the performance of topological photonic sensors, which is indeed a critical factor in assessing the performance of RI sensors, particularly in real-world applications where environmental noise is an important consideration. We have now added a detailed discussion of this critical metric in Sections 2 and 5.2.

Changes in the manuscript:

We added the following paragraph in Section 2 on page 5: "The performance of topological photonic sensors in noisy environments is often quantified by the Signal-to-Noise Ratio (SNR), which measures the ratio of the desired signal (e.g., resonance shift due to a change in refractive index) to the background noise [4]. A high SNR indicates that the sensor can reliably detect small changes in the target parameter, even in the presence of environmental noise. The robustness of these sensors comes from their topologically protected edge states, which are inherently less susceptible to noise from environmental factors such as material imperfections, temperature fluctuations, and vibration. A high Q-factor often achieved in topological photonic sensors leads to a narrower resonance linewidth, increasing the sensor’s ability to resolve small changes in refractive index, thus improving the SNR. To achieve better signal-to-noise ratio (SNR), a cross-polarization technique is applied [5, 6]."

We added the following paragraph in Section 5.2 on page 19: "The robustness of topological photonic sensors against noise is further quantified by their high Signal-to-Noise Ratio (SNR). The topological protection of edge states ensures that light propagates along the material boundaries with minimal scattering, even in the presence of defects or environmental noise. This results in a strong, well-defined signal (e.g., resonance shift) relative to the background noise, leading to high SNR [7, 8]. For example, in RI sensors, the sharp resonance peaks associated with Fano resonances or quasi-bound states in the continuum (quasi-BICs) enable precise detection of small RI changes, even in noisy environments. The high SNR of these sensors makes them particularly suitable for applications in industrial settings or complex biological samples, where environmental noise is often significant. "

Comment4. References are needed in many sections that support the idea that the IR sensors are viable.

Response4:

Thanks for the reviewer's suggestion. The relevant references are beneficial. The references needed are cited in proper positions, for example see Refs with the numbers: 31, 32, 83-88, 92-94, 133, 161-164, and 167.

Comment5. Please describe which molecules are being sensed. Only the possible molecules and gases that can be sensed are mentioned in many sections.

Response5:

We appreciate your comment and acknowledge the need for a more straightforward description of the specific molecules detected by topological photonic sensors. To address this, we have revised the biosensors and gas sensors sections to explicitly mention the specific biomolecules (e.g., cancer biomarkers, SARS-CoV-2 S-glycoprotein, DNA, proteins, viruses, and waterborne bacteria) and specific gases (e.g., NO₂, CH₄, CO₂, and VOCs) that have been experimentally detected using these sensors.

Changes in the manuscript:

We modified the writing in Biosensors and Gas sensor Sections as follows:

  1. In the Biosensors Section, we modified the writing on the first paragraph of section 3.2 on page 8 as: "Topological photonic biosensors enable the precise detection of specific biomolecules, including cancer biomarkers, SARS-CoV-2 S-glycoprotein, DNA fragments, proteins, viruses, and waterborne bacteria, demonstrating their potential for biomedical diagnostics and disease monitoring."
  2. In the Gas Sensors Section, we modified the writing in the first paragraph of section 3.4 on page 11: "These sensors leverage the interaction of light with gaseous media, inducing shifts in resonance or transmission spectra due to RI changes caused by gas adsorption or interaction. Topological photonic gas sensors have been demonstrated to detect nitrogen dioxide (NO₂), methane (CH₄), carbon dioxide (CO₂), and volatile organic compounds (VOCs), making them valuable for environmental monitoring and industrial safety."

Comment6. In section 3.3, line 358, What do you understand by high precision? There is no number to refer to.

Response6:

Thank you for your observation. We acknowledge that “high precision” was used without a specific numerical reference. High precision in thermal sensors refers to the ability to detect and measure small changes in temperature with high accuracy and resolution. For instance, a sensor with a resolution of 0.01°C can detect temperature changes as small as 0.01°C, making it suitable for applications requiring fine temperature control or monitoring.

Changes in the manuscript:

We modified the writing as follows:

Previous version: "The sensor achieves high precision in detecting temperature variations... "

Revised version: "The sensor achieves a sensitivity of 0.085 nm/°C and a quality factor (Q) of 2216.6, enabling precise temperature monitoring in industrial applications."

Comment7. I suggest creating subsections that differentiate theoretical work from experimental.

Response7:

We thank the reviewer for suggesting subsections that differentiate theoretical work from experimental work. While we agree that such an organization could be helpful in some contexts, we believe that the current structure—organized by sensor type (e.g., refractive index sensors, biosensors, gas sensors)—better serves the purpose of this review. Since the experimental work is relatively limited compared to the theoretical discussion, a strict separation into theoretical and experimental subsections may disrupt the logical flow of the review.

We have clearly distinguished between theoretical and experimental work.

Reviewer 2 Report

Comments and Suggestions for Authors

The authors have written a very well-structured and thoughtful review of sensors based on topological states, quasi-bound states in the continuum, and Tamm plasmon polaritons . All chapters are very well thought out and give an idea of ​​the problem even to a reader who does not work in this field of photonics. All modern trends in this area of ​​sensorics and their theoretical basis are very well described. I believe that the review can be published without significant changes. I have only two comments that can be taken into account in the future:

1. Is it possible to increase the resolution of the figures taken from other articles given in the review? It is quite difficult to see all the symbols in these pictures.

2. From the review, I still have one question, namely, which of the directions that arose during the creation of topological sensors described in the review is the most practically feasible? Which of them faces the most significant difficulties in practical implementation? Perhaps this data should be included in the Discussion.

Author Response

General comments:

The authors have written a very well-structured and thoughtful review of sensors based on topological states, quasi-bound states in the continuum, and Tamm plasmon polaritons. All chapters are very well thought out and give an idea of the problem even to a reader who does not work in this field of photonics. All modern trends in this area of sensorics and their theoretical basis are very well described. I believe that the review can be published without significant changes. I have only two comments that can be taken into account in the future

Response:

We greatly thank the reviewer for the high evaluation of our work.

Comment1. Is it possible to increase the resolution of the figures taken from other articles given in the review? It is quite difficult to see all the symbols in these pictures.

Response1:

Thank you for pointing out the figure’s alignment. We increased the resolution figures.

Comment2. From the review, I still have one question, namely, which of the directions that arose during the creation of topological sensors described in the review is the most practically feasible? Which of them faces the most significant difficulties in practical implementation? Perhaps this data should be included in the Discussion.

Response2:
Thank you for your insightful question. We acknowledge the importance of assessing the practical feasibility of different topological sensor types and identifying their key challenges in real-world implementation. In response to your suggestion, we have created a new subsection in Section 5 (Challenges and Future Directions) titled "Practical Feasibility of Topological Photonic Sensors" to provide a detailed discussion on the feasibility and practical limitations of various topological photonic sensors. We discuss the most practically feasible directions (e.g., refractive index and thermal sensors) and the challenges faced by less feasible directions (e.g., biosensors and gas sensors).

Changes in the manuscript:

We added a new subsection, 5.5, titled Practical Feasibility of Topological Photonic Sensors

The practical feasibility of topological photonic sensors varies significantly depending on the sensor type and application. Below, we discuss the most promising directions and their practical implementation challenges.

RI sensors and thermal sensors are among the most practically feasible topological photonic sensors. RI sensors, such as those based on TESs and Fano resonances, have been experimentally demonstrated to have high sensitivity and robustness against noise. For example, the 1D topological photonic crystal heterostructure experimentally reproduces Fano resonance in the optical communication range with a high quality of 104. proposed by Gao, Hu et al. [1] and has been tested in real-world environments. Similarly, thermal sensors leveraging the thermo-optic effect and thermal expansion, such as those proposed by Aly et al. [2], have shown high precision and stability, making them suitable for industrial and environmental monitoring.

Biosensors and gas sensors face more significant challenges in practical implementation. While biosensors based on TPPs and CTCS have demonstrated high sensitivity in detecting biomolecules, their practical application is hindered by the need for functionalized surfaces and the complexity of integrating these sensors into portable devices. Gas sensors face selectivity and environmental stability challenges, particularly those designed to detect trace gases like NO₂ and CO₂. For instance, the graphene-based gas sensor proposed by Janaszek et al. [3] achieved a detection limit of 1 ppb for NO₂ but requires further optimization to minimize cross-sensitivity with other gases and improve long-term stability.

Some emerging sensor technologies, such as non-Hermitian sensors based on EPs and Weyl semimetal-based sensors, remain primarily theoretical. While EP-based sensors can achieve ultra-high sensitivity due to eigenvalue coalescence, their fragile nature and high sensitivity to perturbations make them challenging to implement in practical devices. Similarly, Weyl semimetal sensors, which rely on topological surface states and strong Berry curvature effects, require low-temperature operation, making them impractical for everyday sensing applications. Although these approaches hold promises for future high-precision sensors, they have not yet reached the level of experimental validation necessary for real-world deployment.

Round 2

Reviewer 1 Report

Comments and Suggestions for Authors

I do not have further comments. The mansucript has been improved substantially and suggests its publication.